# v-SNARE transmembrane domains function as catalysts for vesicle fusion

Madhurima Dhara[1†], Antonio Yarzagaray[1†], Mazen Makke[1], Barbara Schindeldecker[1], Yvonne Schwarz[1], Ahmed Shaaban[2], Satyan Sharma[3], Rainer A Böckmann[4], Manfred Lindau[3], Ralf Mohrmann[2*‡], Dieter Bruns[1*‡]

[1]Institute for Physiology, Saarland University, Homburg, Germany; [2]Zentrum für Human- und Molekularbiologie, Saarland University, Homburg, Germany; [3]Group Nanoscale Cell Biology, Max-Planck-Institute for Biophysical Chemistry, Göttingen, Germany; [4]Computational Biology, Department of Biology, Friedrich-Alexander University, Erlangen, Germany

**Abstract** Vesicle fusion is mediated by an assembly of SNARE proteins between opposing membranes, but it is unknown whether transmembrane domains (TMDs) of SNARE proteins serve mechanistic functions that go beyond passive anchoring of the force-generating SNAREpin to the fusing membranes. Here, we show that conformational flexibility of synaptobrevin-2 TMD is essential for efficient $Ca^{2+}$-triggered exocytosis and actively promotes membrane fusion as well as fusion pore expansion. Specifically, the introduction of helix-stabilizing leucine residues within the TMD region spanning the vesicle's outer leaflet strongly impairs exocytosis and decelerates fusion pore dilation. In contrast, increasing the number of helix-destabilizing, ß-branched valine or isoleucine residues within the TMD restores normal secretion but accelerates fusion pore expansion beyond the rate found for the wildtype protein. These observations provide evidence that the synaptobrevin-2 TMD catalyzes the fusion process by its structural flexibility, actively setting the pace of fusion pore expansion.

*For correspondence: ralf. mohrmann@uks.eu (RM); dieter. bruns@uks.eu (DB)

†These authors contributed equally to this work
‡These authors also contributed equally to this work

Competing interests: The authors declare that no competing interests exist.

## Introduction

SNARE-mediated membrane fusion comprises a series of mechanistic steps requiring both protein-protein as well as protein-lipid interactions. Protein-protein interactions involving SNARE proteins in the fusion process have been explored in great detail (*Jahn and Fasshauer, 2012*; *Sudhof and Rothman, 2009*), but the functional role of SNARE-lipid interplay has remained enigmatic. Previous studies provided conflicting views on the requirement of proteinaceous membrane anchors of SNARE proteins for efficient neurotransmitter release or vacuole-vacuole fusion (*Chang et al., 2016*; *Fdez et al., 2010*; *Grote et al., 2000*; *Pieren et al., 2015*; *Rohde et al., 2003*; *Wang et al., 2004*; *Zhou et al., 2013*). Even more unclear is how a proteinaceous TMD may regulate the membrane fusion process. Experiments in reduced model systems have suggested that lipidic SNARE-anchors are inefficient in driving proper fusion between artificial liposomes (*McNew et al., 2000*), cells expressing 'flipped' SNAREs (*Giraudo et al., 2005*), or between liposomes and lipid nanodiscs (*Bao et al., 2015*; *Shi et al., 2012*). However, these experiments were unable to track kinetic intermediates *en route* to fusion (e.g. priming, triggering or fusion pore expansion) leaving the questions unanswered whether and if so, at which step TMDs of SNARE proteins may regulate fast $Ca^{2+}$-triggered exocytosis and membrane fusion (*Fang and Lindau, 2014*; *Langosch et al., 2007*). In comparison to other single-pass transmembrane proteins, SNARE TMDs are characterized by an overrepresentation of ß-branched amino acids (e.g. valine and isoleucine, ~40% of all residues [*Langosch et al., 2001*; *Neumann and Langosch, 2011*]), which renders the helix backbone

**eLife digest** Neurons signal to other cells by releasing chemicals known as neurotransmitters. The neurotransmitters are stored in the neuron in small membrane-bound compartments called vesicles. When a neuron receives an electrical impulse, this ultimately triggers the vesicles to fuse with the cell membrane and release their contents into the gap between the neurons. This process is known as exocytosis. Other cells called neuroendocrine cells, which can receive signals from neurons, also undergo exocytosis to release chemicals into the bloodstream.

A group of membrane-bound proteins called SNAREs help a vesicle to fuse with the cell membrane. SNARE proteins are embedded in both the vesicle and cell membrane, and force them into close proximity. When the two membranes make contact, a small channel called the fusion pore forms and expands to release the vesicle's contents out of the cell.

Synaptobrevin-2 is a SNARE protein found in the vesicle membrane. The part of the protein that sits in the membrane is called the transmembrane domain; however, it is not clear whether this domain plays any role in membrane fusion.

The transmembrane domain of synaptobrevin-2 is rich in certain amino acids that are thought to make it flexible, thereby allowing it to bend and tilt in the membrane. Dhara, Yarzagaray et al. altered these amino acids in such a way that made this domain either more or less flexible than in the normal protein. The results show that in both neurons and a type of neuroendocrine cell called chromaffin cells, exocytosis is significantly reduced and the fusion pores open more slowly when the transmembrane domain is less flexible. By contrast, exocytosis occurs normally when the transmembrane domain is more flexible; however, the fusion pore expands more rapidly than normal.

These results suggest that the flexibility of the transmembrane domain of synaptobrevin-2 promotes membrane fusion and sets the pace at which the fusion pore expands. It is likely that the transmembrane domain disturbs the surrounding membrane in a way that enables these events to happen. Further work is needed to address whether this is the case.

conformationally flexible (*Han et al., 2016*; *Quint et al., 2010*; *Stelzer et al., 2008*). In an α-helix, non-ß-branched residues like leucine can rapidly switch between rotameric states, which favor van der Waals interactions with their i ± 3 and i ± 4 neighbors, thereby forming a scaffold of side chain interactions that defines helix stability (*Lacroix et al., 1998*; *Quint et al., 2010*). Steric restraints acting on the side chains of ß-branched amino acids (like valine and isoleucine) instead favor i ± 4 over i ± 3 interactions leading to local packing deficiencies and backbone flexibility. In vitro experiments have suggested that membrane-inserted short peptides mimicking SNARE TMDs (without a cytoplasmic SNARE motif) exhibit a significant fusion-enhancing effect on synthetic liposomes depending on their content of ß-branched amino acids (*Hofmann et al., 2006*; *Langosch et al., 2001*). Furthermore, simulation studies have shown an inherent propensity of the SNARE TMDs or the viral hemagglutinin fusion peptide to disturb lipid packing, facilitating lipid splay and formation of an initial lipid bridge between opposing membranes (*Kasson et al., 2010*; *Markvoort and Marrink, 2011*; *Risselada et al., 2011*).

Here, we have investigated the functional role of the synaptobrevin-2 (syb2) TMD in $Ca^{2+}$-triggered exocytosis by systematically mutating its core residues (amino acid positions 97–112) to either helix-stabilizing leucines or flexibility–promoting ß-branched isoleucine/valine residues. In a gain-of-function approach TMD mutants were virally expressed in v-SNARE deficient adrenal chromaffin cells (dko cells), which are nearly devoid of exocytosis (*Borisovska et al., 2005*). By using a combination of high resolution electrophysiological methods (membrane capacitance measurements, amperometry) and molecular dynamics simulations, we have characterized the effects of the mutations in order to delineate syb2 TMD functions in membrane fusion. Our results indicate an active, fusion promoting role of the syb2 TMD and suggest that structural flexibility of the N-terminal TMD region catalyzes fusion initiation and fusion pore expansion at the millisecond time scale. Thus, SNARE proteins do not only act as force generators by continuous molecular straining, but also facilitate membrane merger via structural flexibility of their TMDs. The results further pinpoint a hitherto unrecognized

mechanism wherein TMDs of v-SNARE isoforms with a high content of ß-branched amino acids are employed for efficient fusion pore expansion of larger sized vesicles, suggesting a general physiological significance of TMD flexibility in exocytosis.

## Results

### Stabilization of the syb2 TMD helix diminishes synchronous secretion

To study the potential impact of structural flexibility of the syb2 TMD on fast $Ca^{2+}$-dependent exocytosis, we substituted all core residues of the syb2 TMD with either leucine, valine or isoleucine (*Figure 1A*) and measured secretion as membrane capacitance increase in response to photolytic uncaging of intracellular [Ca]i. Replacing the syb2 TMD by a poly-leucine helix (polyL) strongly reduced the ability of the syb2 mutant to rescue secretion in v-SNARE deficient chromaffin cells (*Figure 1B*). Indeed, a detailed kinetic analysis of the capacitance changes revealed that both components of the exocytotic burst, the rapidly releasable pool (RRP) and the slowly releasable pool (SRP), were similarly diminished, and the sustained rate of secretion was reduced, but no changes in exocytosis timing were observed (*Figure 1B*). The similar relative decrease in both, the RRP and the SRP component, could indicate that the polyL mutation interferes with upstream processes like the priming reaction leading to impaired pool formation and reduced exocytosis competence. By studying SNARE complex assembly with recombinant proteins, we found that the polyL variant affects neither the rate nor the extent of SNARE complex formation (*Figure 1—figure supplement 1*). This renders the possibility unlikely that the mutant syb2 TMD allosterically affects the upstream SNARE motif leading to altered interaction with its cognate SNARE partners. Thus, the secretion deficiency in polyL expressing cells is not due to impaired SNARE complex formation, i.e. by causing changes in vesicle priming, but rather reflects defective vesicle fusion.

In contrast, replacing the core residues of the syb2 TMD with either a poly-valine (polyV) or polyisoleucine (polyI) helix resulted in mutants that support exocytosis like the wildtype protein (*Figure 1C,D*). Thus, substitution of a substantial amount of amino acids within the syb2 TMD with either type of ß-branched residue is tolerated without affecting secretion (*Figure 1A,C,D*). Since both, polyV and polyI mutants can functionally replace the wildtype protein, it seems likely that membrane fusion does not critically depend on conserved key residues at specific positions within the syb2 TMD. To substantiate this hypothesis, we substituted single highly conserved TMD amino acids, the $G^{100}L$, or those residues that remain unchanged in the polyV mutant (syb2 $V^{101}A$ and syb2 $V^{112}A$, *Figure 1A*). None of these mutations interfered with the $Ca^{2+}$-triggered secretion response (*Figure 1—figure supplement 2*). Moreover a variant, in which all TMD core residues were substituted by an alternating sequence of leucine and valine (denoted polyLV) in order to match the ~50% ß-branched amino acid content of the syb2 TMD, also rescued secretion like the wildtype protein (*Figure 1E*). In control experiments we further confirmed by epifluorescence and high resolution structured illumination microscopy (SIM) that the syb2 TMD mutant proteins were correctly sorted to chromaffin granules and expressed with similar efficiency as the wildtype protein (*Figure 1—figure supplement 3*).

The strong functional differences seen in $Ca^{2+}$-triggered exocytosis when replacing the TMD core by leucines and isoleucines (or valines, respectively) are remarkable, given that these aliphatic amino acids hardly deviate in their physicochemical properties regarding hydrophobicity (Kyte-Doolittle scale: Leu 3.8, Ile 4.5, Val 4.2) and side chain volume (Leu 168 Å, Ile 169 Å, Val 142 Å). However, an attractive explanation for the different secretory effects of the amino acids is delivered by their different side chain mobility (Leu > Ile/Val), thereby influencing side chain to side chain interactions and TMD back bone dynamics (*Quint et al., 2010*), as will be further explored below (Figure 4).

Taken together, the combined set of mutant phenotypes supports the view that the changes in overall structural flexibility of the TMD, rather than a requirement of specific residues at key positions, determine the exocytotic response by changing vesicle fusogenicity, pointing to an active role of the v-SNARE TMD in membrane fusion.

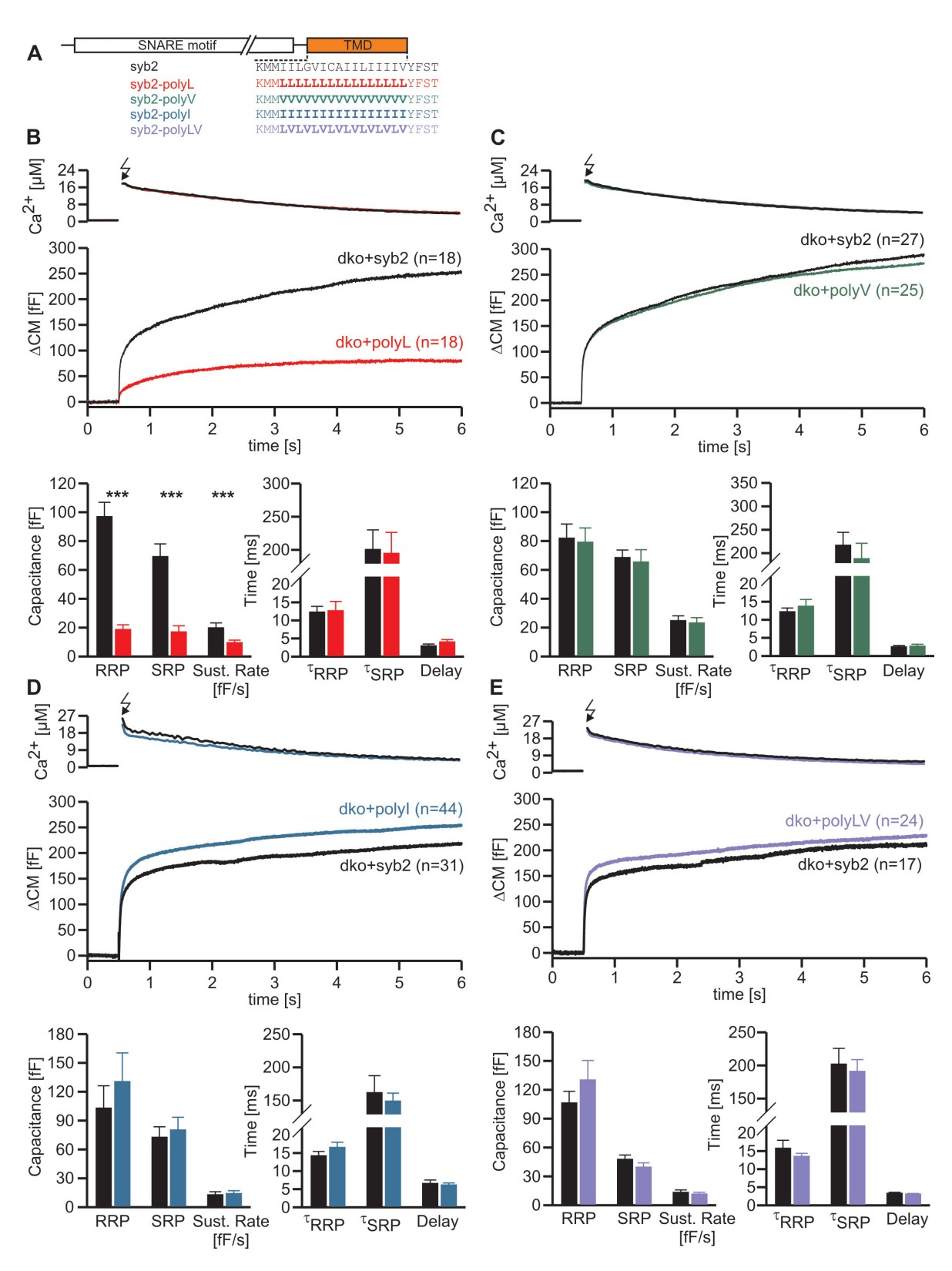

**Figure 1.** Helix stabilizing amino acids in the syb2 TMD diminish secretion. (**A**) Schematic representation of syb2 and corresponding TMD mutants (polyL, polyV, polyI, polyLV). (**B–E**) Mean flash-induced [Ca$^{2+}$]i levels (*top panels*) and corresponding CM responses (*middle panels*) of dko cells

*Figure 1 continued on next page*

*Figure 1 continued*

expressing syb2 wt, polyL, polyV, polyI or polyLV mutants. The polyL mutation reduced RRP and SRP size as well as sustained rate of release, whereas other substitutions of the TMD core residues with valine, isoleucine or a combination of leucine and valine fully restored exocytosis (*bottom panels*). The kinetics of release $\tau_{RRP}$, $\tau_{SRP}$ and the secretory delay are unchanged for all mutants. Arrow indicates flash. Data are represented as mean ± SEM and numbers of cells are indicated within brackets. ***$p < 0.001$, Mann Whitney U test versus syb2.

The following figure supplements are available for figure 1:

**Figure supplement 1.** The poly-L mutant forms SDS-resistant SNARE complexes like the wildtype protein.

**Figure supplement 2.** Substitution of conserved amino acids within the syb2-TMD does not affect vesicle fusion.

**Figure supplement 3.** Syb2 and its TMD mutants are sorted to chromaffin granules with similar efficiency.

## Changing the content of ß-branched amino acids in the syb2 TMD controls the speed of transmitter discharge from single vesicles

Analysis of tonic secretion (evoked by continuous intracellular perfusion with solution containing 19 µM free calcium) with simultaneous membrane capacitance (CM) measurements and carbon fiber amperometry independently confirmed our observations that the polyL mutant diminishes exocytosis, whereas the polyI and polyV variants support secretion at wildtype levels (*Figure 2A–D*). The close correlation between the results of both types of secretion measurements for syb2 and its mutant variants (slope: syb2 0.18 events/fF, $r^2 = 0.97$; polyL 0.17 events/fF, $r^2 = 0.97$; polyV 0.17 events/fF, $r^2 = 0.95$; polyI 0.17 events/fF, $r^2 = 0.94$) shows that the observed CM changes are due to alterations in exocytosis of catecholamine-containing granules. They further render the possibility unlikely that mutant-mediated changes of the CM signal are due to premature closure of the fusion pore and interference with subsequent vesicle endocytosis (*Deak et al., 2004*; *Rajappa et al., 2016*; *Xu et al., 2013*).

Carbon fiber amperometry allows for resolution of discrete phases of transmitter discharge from single vesicles, comprising a prespike signal that reflects transmitter release through the narrow initial fusion pore and a main amperometric spike that coincides with bulk release (*Albillos et al., 1997*; *Bruns and Jahn, 1995*; *Chow et al., 1992*). The polyL variant not only lowered the frequency of exocytotic events but also profoundly slowed transmitter release from the vesicle, compatible with the phenotype of a fusion mutant. The release events were characterized by a decreased amplitude and increased rise-time as well as half-width of the amperometric signal (*Figure 2E,F*). In clear contrast, expression of either polyV or polyI variant accelerated catecholamine release compared to controls, as indicated by significantly higher spike amplitudes, reduced rise-times and half-width values (*Figure 2E,F*). Evidently, modifying the content of ß-branched amino acids within the TMD causes correlated changes in spike waveform, even producing a gain-of-function phenotype in pore expansion kinetics for TMDs enriched in ß-branched residues. Moreover, TMD mutations also affected the prespike signal and its current fluctuations, which report transient changes in neurotransmitter flux through the early fusion pore (*Kesavan et al., 2007*). The polyL mutation prolonged the expansion time of the initial fusion pore, lowered its current amplitude and diminished fluctuations in the signal time-course compared with the wildtype protein (*Figure 3*). The polyV and polyI variants shortened prespike duration, increased its amplitude, and current fluctuations. Taken together, the polyL and poly I/V mutants oppositely affect both, the prespike and the spike phase of transmitter discharge, implying that conformational properties of the syb2 TMD govern the fusion process from the opening of the nascent fusion pore to its final expansion.

## ß-branched residues substantially enhance TMD flexibility

Our mutational analysis suggested that changes in the conformational properties of the TMD can cause characteristic fusion defects, thereby indicating a TMD-based mechanism supporting exocytosis. To further investigate this mechanism we studied the structure and dynamics of the TMD mutants using molecular dynamics simulations of the C-terminal region of syb2 (residues 71–116). Based on the X-ray crystallographic structure (*Stein et al., 2009*), syb2 and its mutant variants were

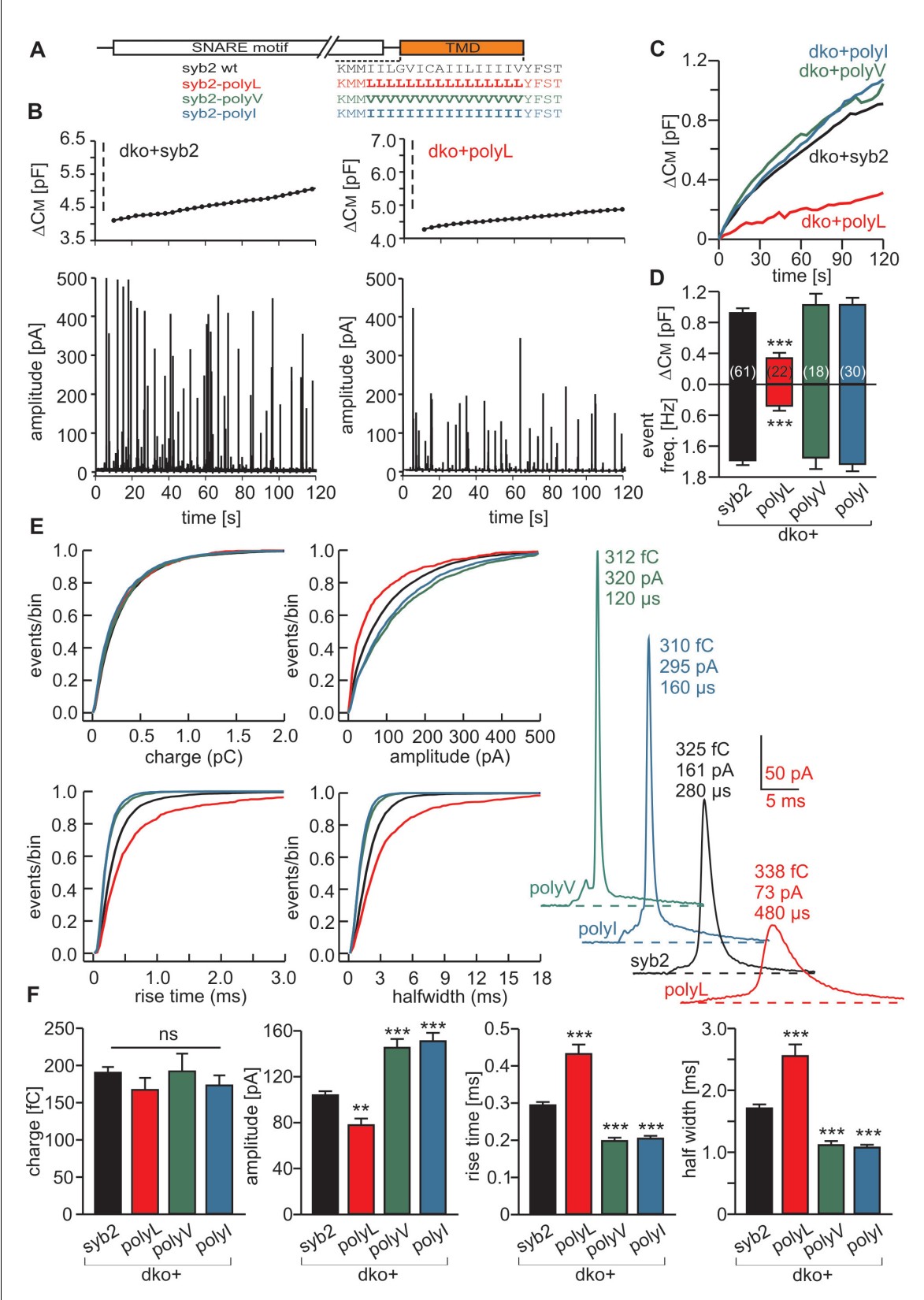

**Figure 2.** Modifying the number of ß-branched residues in the syb2 TMD changes the kinetics of cargo discharge. (**A**) Schematic representation of syb2 and its TMD mutants (polyL, polyV, polyI). (**B**) Exemplary recordings of CM and amperometry for dko cells expressing syb2 or the polyL mutant (dashed

*Figure 2 continued on next page*

*Figure 2 continued*

lines indicate cell opening initiating intracellular perfusion with 19 µM free $Ca^{2+}$). (**C**) Mean capacitance responses over 120 s. (**D**) Total $\Delta CM$ after 120 s (top) and amperometric event frequency (bottom) averaged from the indicated number of cells. (**E**) Properties of the main amperometric spike, displayed as cumulative frequency distribution for the indicated parameters. Exemplary amperometric events with similar charge for the indicated groups are shown (right). (**F**) The polyL mutant prolonged and the polyV or polyI mutations accelerated spike kinetics (causing corresponding changes in spike amplitude) without affecting quantal size. Values are given as mean of median determined from the parameter's frequency distribution for each cell. Data were collected from cells/events measured for syb2 (61/7275), polyL (22/1018), polyV (18/2431), polyI (30/2789). Only cells with >20 events were considered. Data are represented as mean ± SEM. **p<0.01, ***p<0.001, one way analysis of variance versus syb2.

embedded in an asymmetric membrane (mimicking the physiological lipid composition of synaptic vesicles [*Sharma et al., 2015*; *Takamori et al., 2006*]) and structural flexibility was calculated from the root mean square fluctuation (RMSF) of the backbone atoms for each peptide (*Figure 4*). The results show that conformational flexibility of the TMD region is significantly lowered in the polyL and increased in the polyV variant compared with the wildtype protein. Similarly, changes in the root mean square displacement (RMSD) of the $C\alpha$-atoms relative to an ideal $\alpha$-helix (syb2 0.104 ± 0.004 nm; polyL 0.067 ± 0.003 nm, p<0.001; polyV 0.139 ± 0.005 nm, p<0.001, one-way analysis of variance versus syb2) are paralleled by alterations in $\alpha$-helix content of syb2 TMD (79 ± 0.56%) and its variants (polyL 83 ± 0.56%, and polyV 65 ± 5.6%). Taken together, changing the frequency of ß-branched residues within the syb2 TMD substantially varies conformational flexibility, which clearly correlates with alterations in the kinetics of the nascent fusion pore as well as in the spike waveform. Overall, these data provide strong evidence that structural features of v-SNARE TMDs are crucial for $Ca^{2+}$-triggered exocytosis, enabling TMDs to actively promote the fusion process.

## Structural flexibility of the N-terminal region of syb2 TMD catalyzes fusion initiation and fusion pore expansion

Membranes first fuse with their outer leaflets, transiting through a hemifused state, before complete merger (continuity of both leaflets) is reached. To study whether structural flexibility is required throughout the entire TMD region or preferentially in one leaflet, we selectively exchanged either half of the syb2 TMD with leucine residues (*Figure 5A*). For tonic secretion (intracellular perfusion with high $Ca^{2+}$-containing solution, *Figure 5B*) and synchronized exocytosis (photolytic uncaging of intracellular $Ca^{2+}$, *Figure 5—figure supplement 1*), we found that leucine substitution within the N-terminal half of the TMD (amino acid 97–104, polyL-Nt, spanning the outer leaflet of the vesicle membrane) failed to fully rescue exocytosis. A similar replacement of amino acids in the corresponding C-terminal half (amino acids 105–112, polyL-Ct, spanning the inner leaflet of the vesicle membrane) was without any effect when compared with the wildtype protein (*Figure 5B* and *Figure 5—figure supplement 1*). Furthermore, an exchange of the N-terminal amino acids with ß-branched isoleucines (polyI-Nt) rescued secretion like the wildtype protein.

Detailed analysis of amperometric events with respect to spike (*Figure 5C–E*) and prespike properties (*Figure 5—figure supplement 2*) showed that exchanging the N-terminal half of the syb2 TMD with either leucine or isoleucine sufficed to reproduce the altered fusion pore behavior seen with an overall exchange of the TMD residues (compare *Figures 2* and *5*). For parameters describing the main spike kinetics we found a clear proportionality between the speed of catecholamine discharge and the number of ß-branched residues near the N-terminal end of the TMD helix (rise-time, $r^2 = 0.95$; half-width $r^2 = 0.96$, *Figure 6A,B*). Moreover, the pre-spike duration is progressively shortened by increasing the fraction of ß-branched residue in the N-terminal half of the TMD ($r^2 = 0.93$) (*Figure 6C*). Following the same line, we also found a strong correlation between the frequency of pre-spike current fluctuations of the mutant variants and their valine/isoleucine-content ($r^2 = 0.96$). Evidently, increasing or decreasing the number of ß-branched amino acids in the TMD oppositely controls conformational flexibility in the TMD helix (*Figure 4*) and the rate of cargo release from single vesicle. These findings provide strong evidence for a mechanistic link between TMD flexibility and the kinetics of fusion pore expansion. However, deviating from this pattern, total secretion in $Ca^{2+}$-infusion and $Ca^{2+}$-uncaging experiments was not further potentiated beyond the wildtype response by increasing the fraction of ß-branched residues within the N-terminal half of the TMD

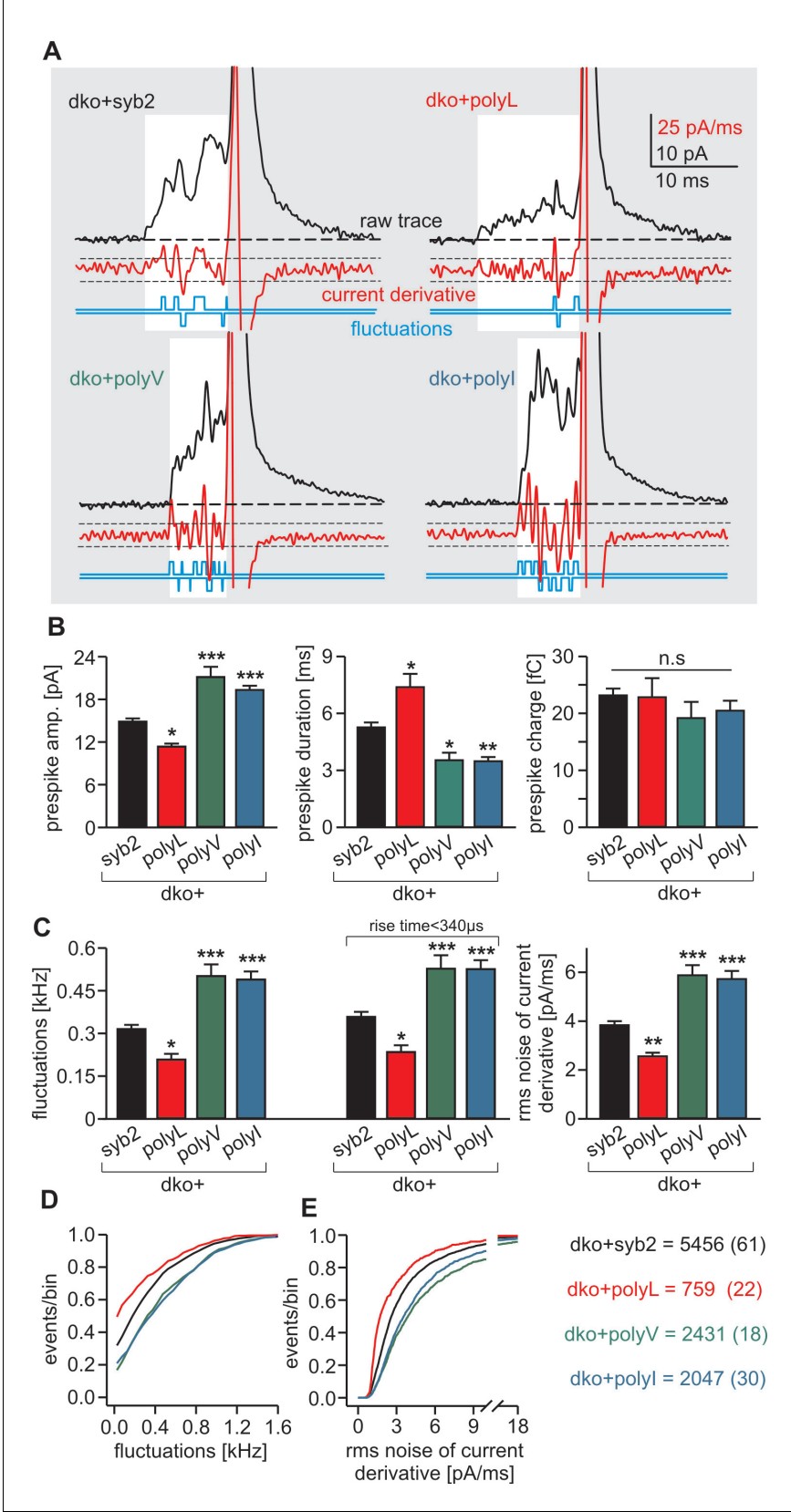

**Figure 3.** PolyL and polyV (or I) mutations oppositely alter the kinetics of prespike signals. (**A**) Exemplary prespike events and analysis of their current fluctuations (highlighted area) during transmitter discharge through a narrow

*Figure 3 continued on next page*

*Figure 3 continued*

pore. Deflections of the current derivative (red trace) above the threshold (dashed lines = ± 4 SD of base line noise) were counted as fluctuations (blue trace). The displayed events have a similar total charge and 50%–90% rise time (dko+syb2: 477fC, 280 μs; dko+polyL: 458fC, 240 μs; dko+polyV: 462fC, 200 μs; dko+polyI: 483fC, 200 μs), indicating that the different fluctuation behavior is not due to differences in diffusional broadening of the current signals. (B) PolyL mutation and polyV or I mutations oppositely altered the amplitude and kinetics of the prespike event, without changing its charge. (C) The average fluctuation frequency (sum of positive and negative fluctuations) of all events with an amplitude >7 pA as well as of events with spike rise times <340 ms (minimizing a potential distortion of the signal time course by diffusional broadening) decreased for the polyL mutant and increased for the polyV/I mutants. Mean rms noise of the current derivative during the prespike signal, serving as threshold-independent parameter of fusion pore jitter, confirms mutant protein-mediated changes in fusion pore dynamics. (D–E) Cumulative frequency distributions for fluctuation frequency and rms noise of current derivative from dko+syb2 (black), dko+polyL (red), dko+polyV (green) and dko+polyI cells (blue). Data were collected from the indicated number of events (cells) and are represented as mean ± SEM. *p<0.05, **p<0.01, ***p<0.001, one way analysis of variance versus syb2.

(*Figure 6D*). Most likely, docking and priming reactions become rate-limiting (*Sorensen, 2009*), thereby preventing the total release to exceed wildtype levels.

Taken together, the syb2 TMD possesses an inherent functional polarity, with the N-terminal region being more important for fusogenicity than the C-terminal side. These observations agree well with previous coarse-grained models of SNARE-mediated fusion events (*Risselada et al., 2011*), suggesting a similar directionality of SNARE TMDs in perturbing lipid packing (enhancing lipid

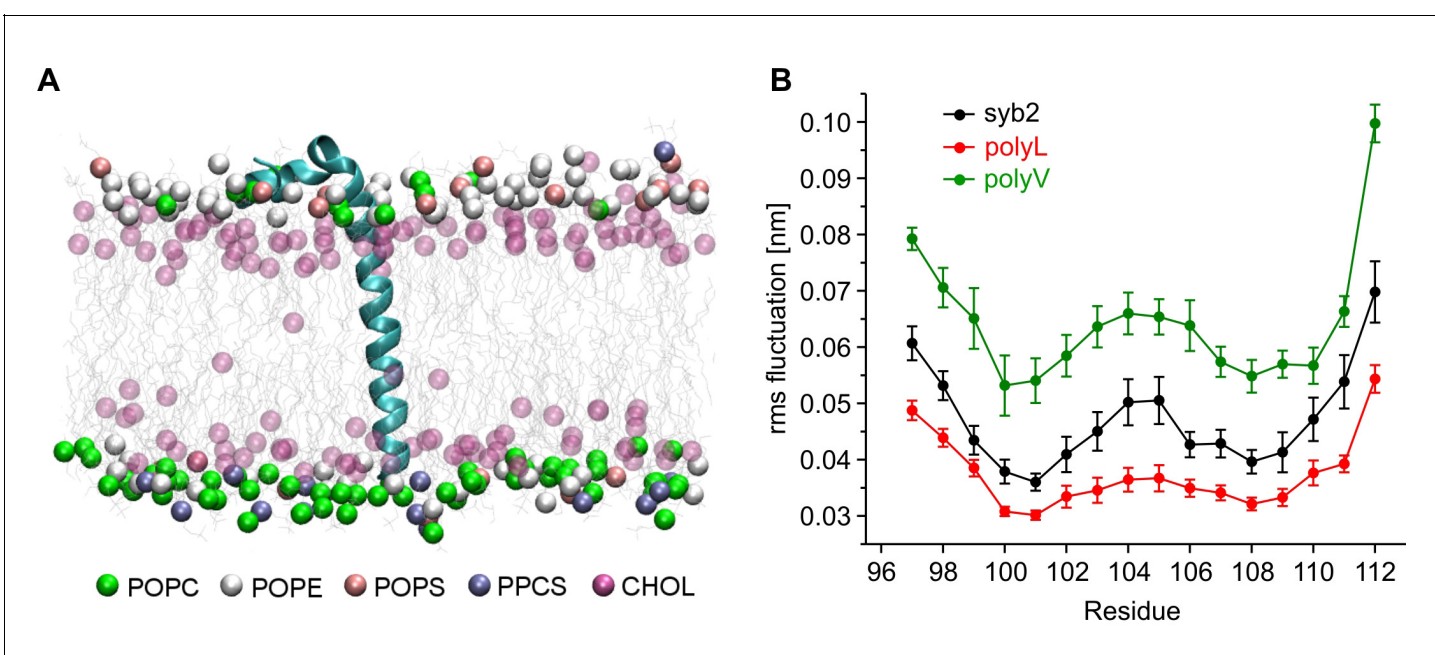

**Figure 4.** Conformational flexibilities of syb2 and its mutant variants. (A) Snapshot from the atomistic simulation for inserted syb2 (residues 71–116) in an asymmetric, self-assembled membrane (cytoplasmic leaflet, top; intravesicular leaflet, bottom) with the protein backbone depicted in cartoon representation. The phosphate atoms of lipid and the hydroxyl carbon of cholesterol are shown in the Van der Waals representation. Other atoms of the lipids are shown as grey lines (water molecules are not shown for clarity). Different lipid moieties are depicted according to the colour code shown below. Note the asymmetric lipid composition of the bilayer. (B) Root mean square fluctuations (RMSF) of C-α atoms derived from 40 ns simulation runs for syb2 and the mutants relative to the average structure of the corresponding peptide. Flexibility of the TMD region (residues 97–112) was determined from non-overlapping 10 ns periods during the last 30 ns of the trajectories for each simulation. The polyV region of the mutant showed on average an increased flexibility (0.0644 ± 0.0029 nm, p<0.001), while the polyL region (0.0374 ± 0.0017 nm, p<0.05) showed a decreased jitter when compared with syb2 (0.0472 ± 0.0022 nm, one-way analysis of variance versus syb2). Error bars are s.e.m. of averages calculated from the values of the individual 10 ns windows (n = 9 for wt, n = 6 for mutants).

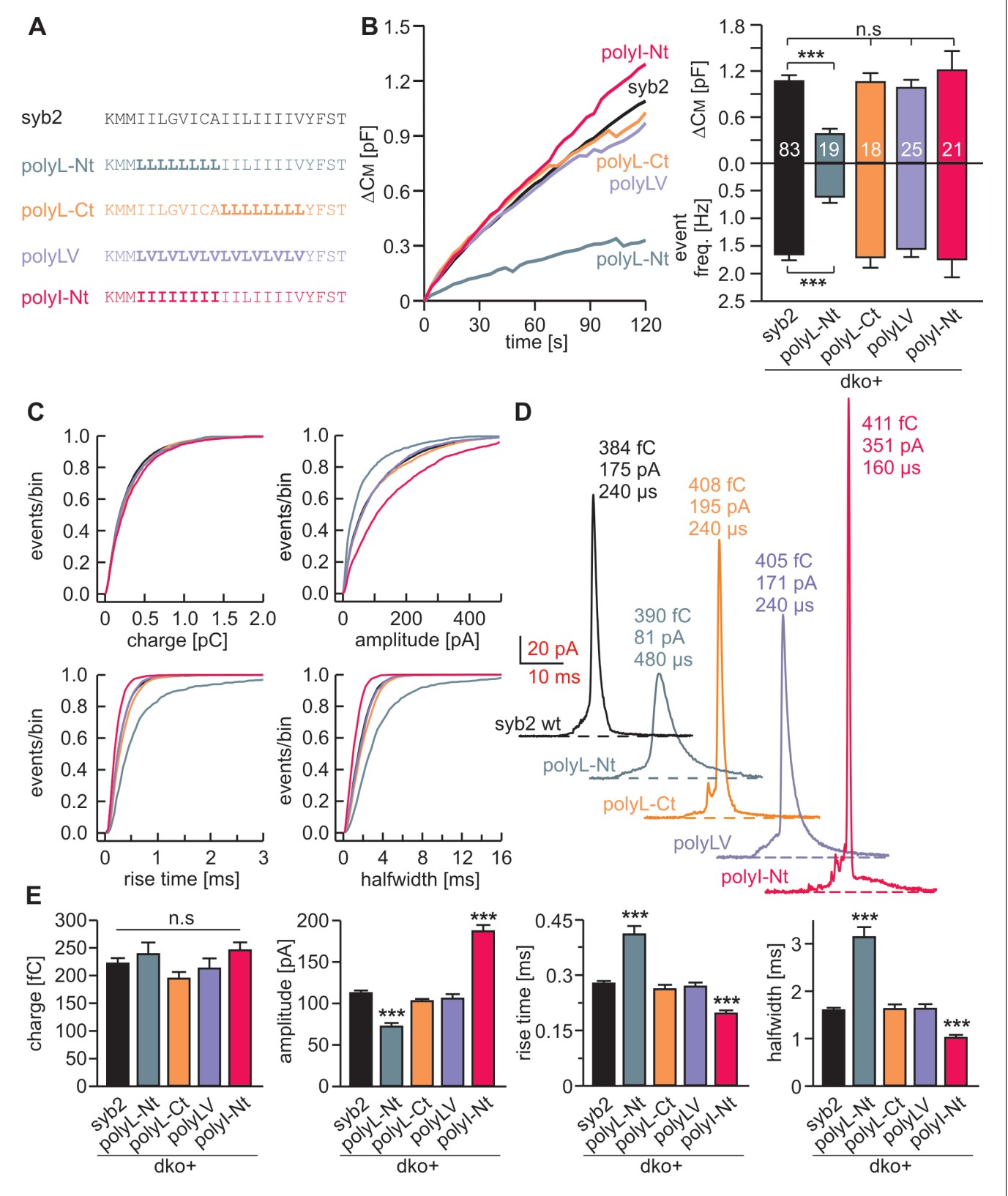

**Figure 5.** Structural flexibility of the N-terminal TMD region catalyzes fusion initiation and fusion pore dilation. (**A**) Schemes of syb2 and corresponding TMD mutants (polyL-Nt, polyL-Ct, polyLV, polyI-Nt). (**B**) Mean capacitance changes in response to intracellular perfusion with 19 μM free $Ca^{2+}$ in the indicated groups. Total ΔCM (top) and amperometric event frequency (bottom) measured over 120 s show that only polyL-Nt mutant fails to rescue

*Figure 5 continued on next page*

*Figure 5 continued*

normal exocytosis. Data are averaged from the indicated number of cells. (**C**) Properties of the main amperometric spikes displayed as cumulative frequency distribution for the indicated parameters and color coded according to (**A**). (**D**) Exemplary amperometric events with similar charge but altered release profile for the indicated syb2 variant. (**E**) PolyL-Nt mutation slowed the spike waveform (reduced amplitude, increased rise time, and half width) while polyI-Nt increased the amplitude and decreased the rise time and half width. Values are given as mean of median determined from the indicated parameter's frequency distribution for each cell. Data were collected from cells/events measured for syb2 (83/9054), polyL-Nt (19/951), polyL-Ct (18/2684), polyLV (25/3576), polyI-Nt (21/2057). Only cells with >20 events were considered. Data are represented as mean ± SEM. ***p<0.001, one-way analysis of variance versus syb2.

The following figure supplements are available for figure 5:

**Figure supplement 1.** Conformational flexibility of the N-terminal region of syb2 TMD supports LDCV fusion.

**Figure supplement 2.** The N-terminal region of syb2 TMD controls kinetics and fluctuations of the early fusion pore.

splaying) preferentially in the cytoplasmic leaflets and, thereby, facilitating the first hydrophobic encounter for forming a lipid bridge between opposing membranes (*Figure 6E*). Similarly, TMD backbone dynamics within the outer leaflet of the fusion pore neck may lower its high membrane curvature, driving fusion pore expansion (*Figure 6E*).

## Lipid anchoring of syb2 aggravates fusion incompetence

A partial rescue of synaptic transmission has previously been observed in cortical syb2$^{-/-}$ neurons expressing an acylated syb2-CSP fusion protein lacking the TMD (*Zhou et al., 2013*). This finding has been interpreted as evidence that v-SNARE TMDs are functionally interchangeable with lipidic membrane anchors. Opposing this view, a recent study showed that the same lipid-anchored syb2 provides little support for spontaneous synaptic transmission (*Chang et al., 2016*). We also found that this acylated syb2-CSP fusion protein was largely inefficient in reconstituting Ca$^{2+}$-triggered exocytosis in chromaffin cells (21% of syb2, *Figure 7A,B*), albeit showing similar expression levels and sorting to granules as the wildtype protein (*Figure 7—figure supplement 1*). Interestingly, while expression of syb2-CSP raises secretion significantly over the level of the dko (2% of syb2), the phenotype is still more severe than the secretion deficits seen with the polyL variant (35% of syb2, *Figure 2C,D*), reconfirming that the proteinaceous membrane anchor provides an autonomous facilitating function in Ca$^{2+}$-triggered exocytosis. Furthermore, like the polyL mutant, the lipid-anchored syb2 prolonged the time course of transmitter discharge during the spike phase (without changing the event charge) (*Figure 7C*) and even more strongly slowed down kinetics of the early fusion pore (*Figure 7D–F*). Collectively, these results highlight the important role of the proteinaceous syb2 membrane anchor in membrane fusion, generally facilitating fusion initiation and pore expansion. Our data obtained with the acylated syb2-CSP fusion protein appear to deviate from the previously reported results by *Zhou et al. (2013)*, wherein the mutant protein significantly rescued synaptic transmission compared to a syb2-RST-mVenus construct serving as control. However, as reported by *Chang et al. (2016)*, the syb2-RST-mVenus construct does not support wildtype like fusion both, in neurons and neuroendocrine cells due to the presence of a positively charged arginine residue in the C-terminal end of the syb2 TMD. Indeed, a previous study has shown that insertion of charged residues in the C-terminal end of the syb2 TMD impairs the release response (*Ngatchou et al., 2010*). Consequently, the reduced ability of the syb2-RST-mVenus construct to rescue neuronal exocytosis may have led to an overestimation of the acylated syb2-CSP response providing an explanation for the apparently discrepant results.

In any case, since SNARE-mediated fusion of SSVs might mechanistically deviate from granule secretion in neuroendocrine cells, we also analyzed the impact of our syb2 TMD mutants on fast glutamatergic release in autaptic cultures of syb2$^{-/-}$ hippocampal neurons. Viral expression of the polyV mutant rescued evoked synaptic transmission to the level of wildtype cells, while the polyL mutant largely failed to support neurotransmitter release (*Figure 7—figure supplement 2A–D*), which is reminiscent of our findings in neuroendocrine cells. Immunofluorescence analyses confirmed that polyL and polyV mutant proteins were indeed targeted to synaptic vesicles with comparable efficiency as the wildtype protein (*Figure 7—figure supplement 2E–G*). To test whether the flexibility

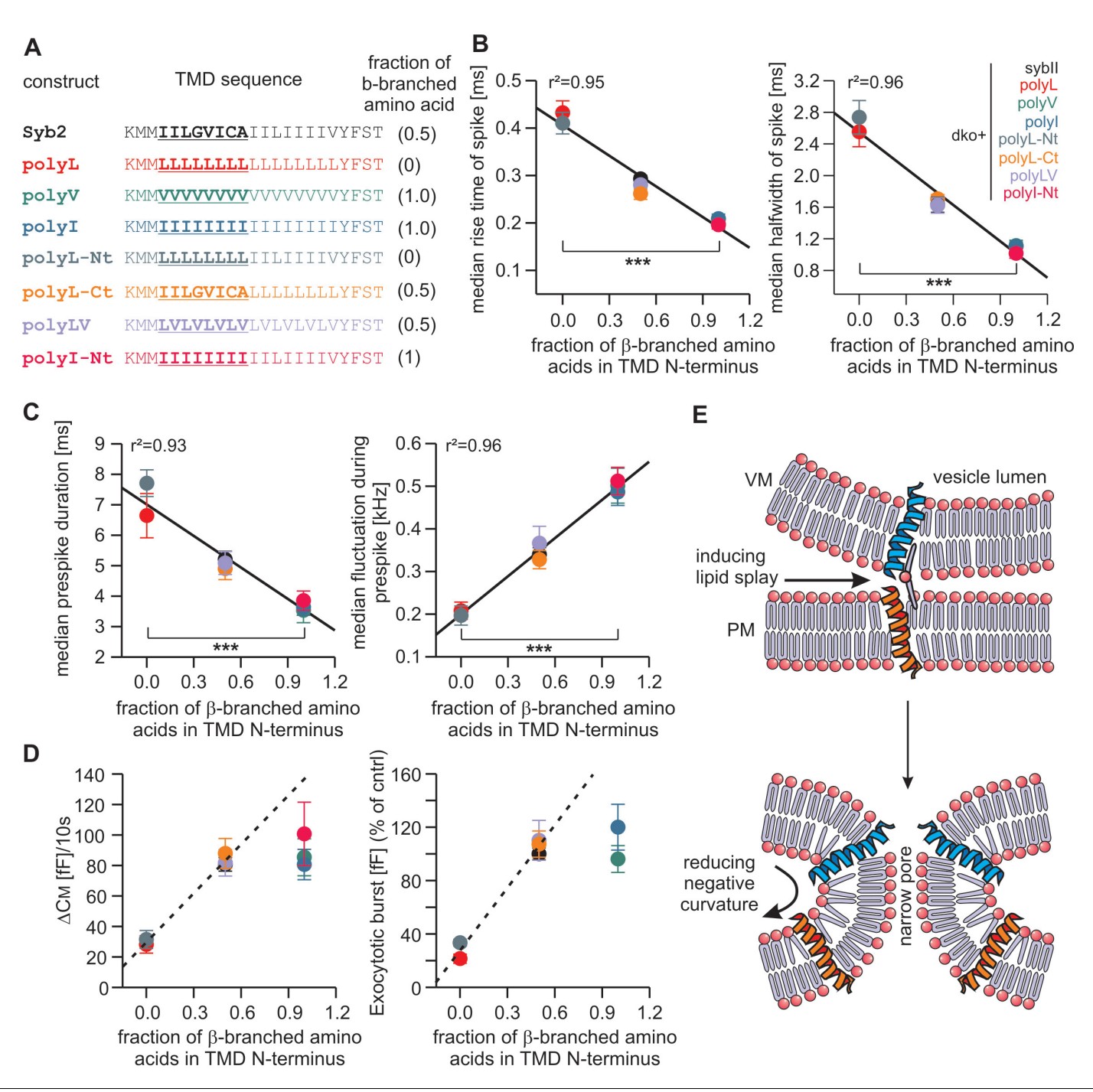

**Figure 6.** Speed of cargo release is systematically correlated with the number of β-branched amino acids in the N-terminal region of the syb2 TMD. (**A**) Schemes of syb2 and corresponding mutants depicting the fraction of β-branched amino acids in the N-terminal region of the TMD (underlined). (**B–C**) Increasing the fraction of β-branched amino acids accelerates the rate of cargo release (spike) as well as the dynamics of the nascent fusion pore (prespike). (**D**) Tonic and synchronous secretion are reduced with the loss of ß-branched amino acids but cannot be further potentiated by enriching ß-branched amino acids in the TMD N-terminal region when compared with syb2. (**E**) Hypothetical models illustrating how conformational flexibility of the syb2 TMD (specifically of the N-terminal region) enhances lipid splay to promote intermembrane contact (PM, plasma membrane, VM, vesicle membrane) during fusion initiation and lowers negative membrane curvature (outer leaflet) to facilitate pore expansion. Data are represented as mean ± SEM. ***p<0.001, one-way analysis of variance between indicated groups.

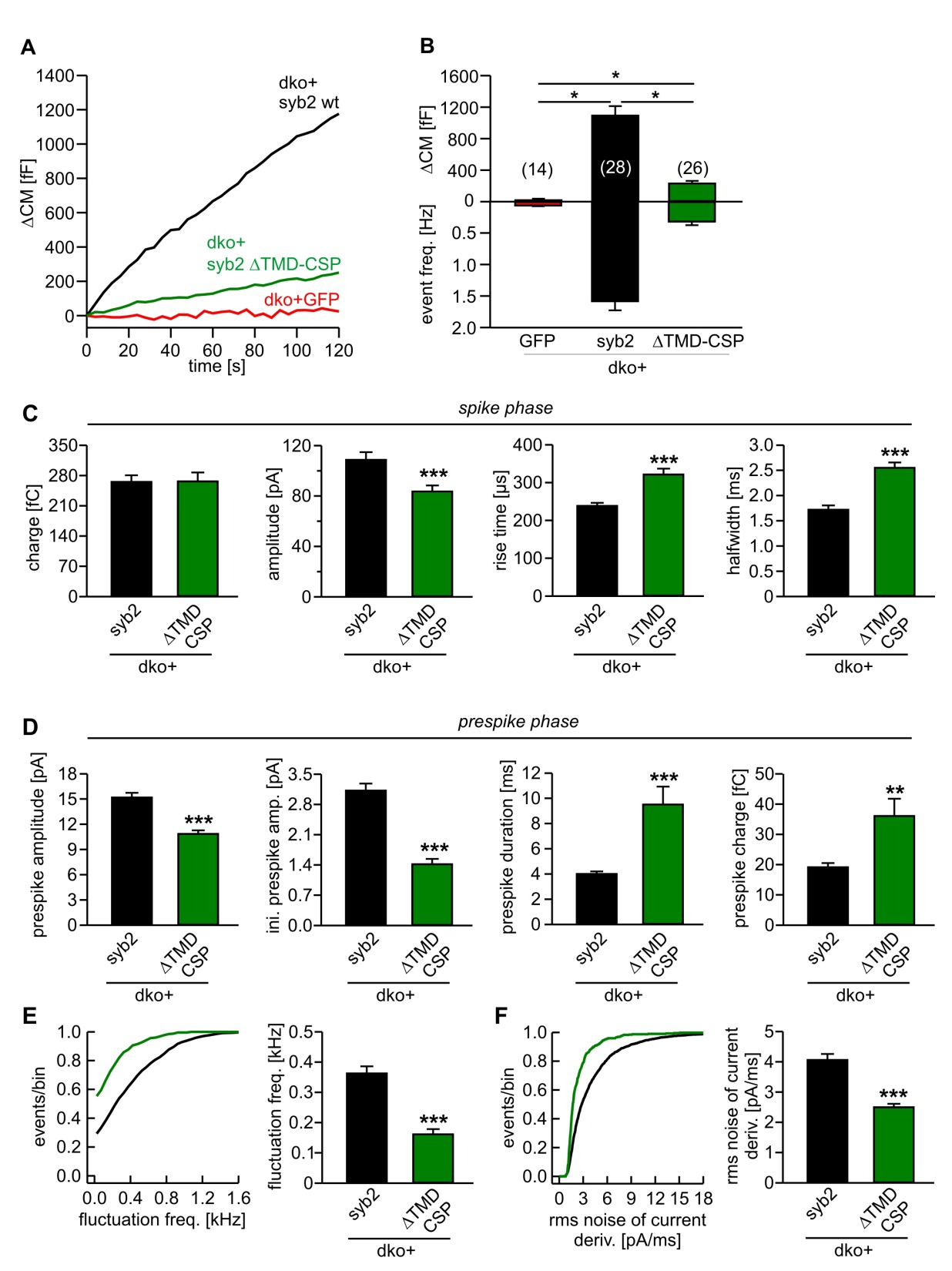

**Figure 7.** Lipid anchored syb2 failed to support normal secretion from chromaffin cells. (**A**) Mean capacitance responses upon intracellular perfusion with 19 μM free $Ca^{2+}$ in the indicated groups. (**B**) Total ΔCM as well as amperometric event frequency measured over 120 s show that the lipid-

*Figure 7 continued on next page*

*Figure 7 continued*

anchored syb2 (ΔTMD-CSP) restored secretion above levels of dko cells but largely failed to support exocytosis like the wildtype protein indicating the functional necessity of a proteinaceous membrane anchor for unperturbed fusion. Data are averaged from the indicated number of cells. ANOVA followed by Kruskal-Wallis post hoc test was performed. (C) Properties of the amperometric spike phase, displayed as cell weighted averages show that the ΔTMD-CSP mutant decreased the spike amplitude while increasing the spike rise time and half width. (D) Effects of the ΔTMD-CSP mutation on the indicated prespike parameters. (E–F) Prespike fluctuations and rms noise of the current derivative are significantly reduced compared to control. Data were collected from events (cells): dko+syb2 4267 (36); dko+ΔTMD-CSP 1078 (34) and are represented as mean ± SEM. ***p<0.001, Mann Whitney U test versus syb2.

The following figure supplements are available for figure 7:

**Figure supplement 1.** Lipid-anchored syb2 ΔTMD-CSP mutant exhibit similar expression and sorting to granules like the wildtype protein.

**Figure supplement 2.** β–branched amino acids in the syb2 TMD regulate synaptic vesicle fusion.

of the syb2 TMD is also critical for quantal signaling, we recorded spontaneous excitatory postsynaptic currents (mEPSCs) in the presence of 1 µM TTX using mass cultures of hippocampal neurons. Compared with the wildtype syb2 protein, expression of the polyL mutant in syb2 ko neurons significantly reduced the frequency of spontaneous events, whereas the polyV mutant fully rescued spontaneous release (ko+syb2: 1.46 ± 0.24 Hz, n = 52; ko+polyL: 0.49 ± 0.12 Hz, n = 23; ko+polyV: 1.42 ± 0.31, n = 21). Notably, the frequency of mEPSCs recorded for the polyL mutant is more than 20fold higher compared to syb2 ko neurons (0.02 ± 0.005 Hz, n = 8), emphasizing the gain-of-function phenotype of the TMD mutant. In contrast, we failed to detect significant alterations in the mean amplitude of the polyL or polyV-mediated mEPSCs compared with the wildtype controls. Potential changes in the release profile of small synaptic vesicles (SSVs) may be masked by dendritic filtering of the receptor-mediated response. Moreover, release from SSVs might be less dependent on TMD-mediated acceleration of fusion pore expansion due to the high curvature of the vesicle, as will be discussed below.

Taken together, comparable deficits in exocytosis are observed for the TMD mutants in neurons as well as neuroendocrine cells indicating similar structural requirements for v-SNARE TMDs to initiate fusion.

## Discussion

The membrane-bridging interactions of SNARE proteins bring vesicle and plasma membrane into close apposition and mediate membrane fusion, but the mechanistic events leading to the formation and control of the exocytotic fusion pore have remained unknown. Here, we studied whether the syb2 TMD serves mechanistic functions beyond the passive membrane-anchoring of the force-generating SNARE complex. By systematically changing the structural flexibility of the syb2 TMD (*Figure 4*), we observed secretion phenotypes that highlight the functional impact of the TMD at various stages of membrane fusion. Our experiments provide first evidence that the syb2 TMD plays an active role in Ca$^{2+}$-triggered exocytosis, acting as a crucial catalyst for membrane merger and fusion pore. These observations raise the important question how TMDs actually contribute to the fusion mechanism.

While the TMD variants tested here leave the stimulus-secretion coupling unchanged (*Figure 1*), mutant syb2 variants designed to dissipate the force transfer between SNARE motif and TMD have been reported to clearly prolong the exocytotic delay (*Kesavan et al., 2007*), providing independent evidence for a distinctive and autonomous function of the TMD in membrane fusion. Neither the overall expression level nor colocalization analyses with the intrinsic marker protein cellubrevin delivered evidence for inefficient sorting of the different mutant proteins to chromaffin granules, thus attributing potential fusion deficits to changes in TMD-mediated function (*Figure 1—figure supplement 3*). An exciting interpretation of our data is that TMD mutations may change protein-lipid interactions during Ca$^{2+}$-dependent fusion by altering the conformational dynamics of the helical backbone. Indeed, previous NMR-studies have shown that increasing the content of ß-branched amino acids of TMD mimic peptides profoundly enhanced lipid mobility and wobbling of lipid head

groups (*Agrawal et al., 2010*, *2007*). Hydrophobic nucleation events, in which lipid tails from opposite membranes initially interconnect the adjacent leaflets, have been identified as a highly energy-demanding step *en route* to fusion (*Kasson et al., 2010*; *Risselada et al., 2011*; *Smirnova et al., 2010*). The reduced fusogenicity of vesicles in chromaffin cells expressing either rigid polyL or lipid anchored syb2 variants indicates that the interplay between flexible SNARE TMDs and surrounding lipids could promote hydrophobic tail protrusion and thereby fusion initiation (*Figure 6*). Indeed, our results are supported by previous in vitro work, showing that isolated SNARE TMDs (*Langosch et al., 2001*) or syb2-juxtamemembrane region-TMD constructs (*Tarafdar et al., 2015*) facilitate liposome-liposome fusion. A similar dependence on protein-lipid interactions for membrane fusion has previously been observed with viral fusogens (*Kasson et al., 2010*; *Tamm et al., 2003*), suggesting that $Ca^{2+}$-triggered exocytosis and viral fusion engage common mechanisms to drive membrane fusion. Given that the formation of an initial lipid stalk is generally observed in direct vicinity of SNARE TMDs in MD simulations (*Risselada et al., 2011*), TMD-rigidifying mutations (e.g. polyL mutation) may lower the probability of lipid splay and thereby produce more unsuccessful fusion attempts with vesicles arrested in a trapped state prior to membrane merger. As fusion mutants are usually expected to slow down stimulus secretion coupling, a scenario where vesicles are led into a trapped state would explain why helix-rigidifying mutations do not alter the kinetics of the exocytotic burst component (*Figure 1*). Regardless of the exact underlying mechanism, our results support a model wherein conformational flexibility of a proteinaceous v-SNARE TMD is required to surmount the energy barrier for initial membrane merger.

Fast and efficient discharge of bulky cargo molecules from large secretory vesicles is bound to expansion of the exocytotic fusion pore. Since bilayer bending mechanics allow pores of smaller vesicles to expand more rapidly (*Alvarez de Toledo et al., 1993*; *Chizmadzhev et al., 1995*; *Zhang and Jackson, 2010*), increasing the content of ß-branched amino acids in the v-SNARE TMD can ease the expansion of a lipidic pore for larger vesicles as they fuse. Our results show that systematically changing the number of helix-destabilizing, ß-branched valine or isoleucine residues in the syb2 TMD leads to correlated changes in fusion pore behavior. In particular, increasing the number of ß-branched residues within the N-terminal half of the TMD causes an unprecedented gain-of-function phenotype, wherein fusion pore dilation is even accelerated beyond the rate found for the wild-type protein, emphasizing the key role of structural dynamics of the syb2 TMD in membrane fusion. Both, substitution of the syb2 TMD with a lipid anchor or with rigidifying leucine residues strongly slowed down kinetics of transmitter discharge, demonstrating the inherent propensity of the syb2 TMD to promote fusion pore expansion. An attractive explanation for this phenomenon could be that structural flexibility within the N-terminal half of the syb2 TMD counters the highly negative curvature of the membrane's outer leaflet to drive expansion of the narrow fusion pore neck (*Figure 6E*). Similarly, $Ca^{2+}$-bound synaptotagmin-1 (syt1) induces positive curvature to the cytoplasmic leaflets of the fusing membranes (*Hui et al., 2009*; *Martens et al., 2007*) and thereby may destabilize the early fusion pore (*Dhara et al., 2014*). In this context, it stands to reason that SNARE force-mediated membrane straining (*Kozlov et al., 2010*) and TMD-mediated lipid perturbation together with syt1's ability to bend membranes are synergistic mechanisms that provide mutual reinforcement to form a nascent lipid bridge between membranes and to drive subsequent pore expansion.

As an alternative hypothesis, membrane-spanning v- and t-SNARE TMDs have been proposed to form channel structures that are aligned in a stacked manner to generate a gap junction-like pore through the vesicular membrane and plasma membrane (*Bao et al., 2015*; *Chang et al., 2015*; *Han and Jackson, 2005*). However, this concept of a proteinaceous fusion pore is difficult to reconcile with our observation that an acylated syb2-CSP fusion protein lacking the TMD can still significantly raise secretion over dko levels (*Figure 7A,B*). Furthermore, TMD variants furnishing hydrophobic, identical residues can rescue (polyLV, *Figure 5* and *Figure 5—figure supplement 2*) or even speed up (polyV and polyI) transmitter discharge, albeit these helices neither exhibit any polarity nor asymmetry with respect to the side-chain volume of residues that could generate different surfaces of the putative proteinaceous pore.

Homotypic TMD-TMD interactions have been implicated in fusion between vacuoles (*Hofmann et al., 2006*) and may be involved in a supramolecular assembly of SNARE proteins that precedes the hemifusion state along the fusion pathway (*Lu et al., 2008*). However, considering the phenotypes within our set of different TMD mutants (G100L, polyV, polyI, polyLV, polyL-Ct, polyL-

Nt), we found that neither key residues for syb2 TMD dimerization (G$^{100}$, *Figure 1—figure supplement 2*) (*Fdez et al., 2010*), nor those that comprise the interacting helical face of the TMDs (L$^{99}$, C$^{103}$, I$^{106}$, I$^{110}$, [*Laage and Langosch, 1997*; *Roy et al., 2004*; *Tong et al., 2009*]), play a significant role for membrane fusion or fusion pore expansion.

In this context, it is noteworthy that neither deletion nor substitutions of membrane-proximal tryptophane (Trp) residues within the juxtamembrane domain (JMD) of syb2 (*Borisovska et al., 2012*) were found to alter the tonic secretion response or fusion pore properties as observed here with the TMD mutants (*Figure 5*). Thus, it is unlikely that TMD mutants interfere with functions of the Trp moiety which influences the electrostatic surface potential by controlling the JMD position at the membrane-water interface (*Borisovska et al., 2012*).

Moreover, the fully zippered cis-SNARE complex (all SNAREs in one membrane) also establishes several stabilizing interactions between the TMDs of syb2 (I98, L99, I102, I106) and syntaxin-1 (syx1) (*Stein et al., 2009*) that might be compromised by mutating the TMD core residues. Yet, several lines of evidence render the possibility unlikely that such a scenario is responsible for functional deficits observed with the TMD mutants. First, the complete substitution of syx interacting residues in syb2 TMD with valine (or isoleucine) had no effect on the total secretion and even accelerated fusion pore expansion compared with the wildtype protein. Secondly, the polyV (*Figure 2*) and the polyLV (*Figure 5*) mutants exhibited different kinetics of fusion pore expansion, even though the crucial amino acids I98, I102 and I106 were substituted by valine residues in both mutant variants. These results counter the view that perturbations of 'lock and key' like protein-protein interactions between syb and syx TMD are responsible for the functional effects of the TMD mutants. Third, neither short insertions of amino acids (e.g. 2 residue KL insertion) nor insertion of 2 helix breaking proline residues, immediately upstream of the syb2 TMD (*Kesavan et al., 2007*), which should interfere with N- to C-terminal zipping of SNAREs into the bilayer spanning helical bundle, were found to affect overall secretion or fusion pore properties. Even a 5 amino acid insertion had no functional consequences on fusion pore dynamics (*Kesavan et al., 2007*). These results together with the strong fusion deficits observed for polyL and polyL-Nt mutants suggest thatconformational flexibility of the syb2 TMD (within the cytoplasmic leaflet of the membrane) rather than defined protein-protein interactions upon progressive zipping of syb2/syx TMDs facilitates secretion and fusion pore expansion. Thus, heterodimerization between v- and t-SNARE TMDs likely succeeds but does not promote fusion pore opening and expansion. Notably, single point mutations (G100L, V101A, V112A, *Figure 1—figure supplement 2*), may similarly change structural flexibility of the TMD. Yet, given the observed proportionality between the number of ß-branched amino acids and fusion pore parameters (*Figure 6*), they are not expected to detectably affect fusion pore dynamics.

The increasing energy barrier for larger vesicles to overcome bilayer bending within their nascent fusion pores is documented in amperometric recordings, showing that larger vesicles form more stable initial fusion pores (i.e. longer prespike duration, [*Alvarez de Toledo et al., 1993*; *Chizmadzhev et al., 1995*; *Zhang and Jackson, 2010*]). The observed systematic dependency of fusion pore dynamics on the number of ß-branched amino acids in the syb2 TMD raises the question whether structural flexibility of TMDs indeed varies among other v-SNARE isoforms and thus could facilitate cargo release in the context of diverse physiological processes. Interestingly, v-SNARE isoforms responsible for exocytosis of differentially-sized secretory vesicles show a considerable degree of variability regarding the content of ß-branched amino acids within the N-terminal half of their TMDs (*Table 1*). v-SNARE proteins, like VAMP7 and VAMP8, contain more than 70% ß-branched amino acids within this TMD region and thereby are well-suited for exocytosis of large zymogen granules and mast cell vesicles facilitating rapid pore expansion and release of their bulky cargo molecules such as interferon (*Krzewski et al., 2011*) and hexoaminidase (*Lippert et al., 2007*; *Wang et al., 2004*). Others, like cellubrevin (VAMP3) or syb2 (VAMP2), with an intermediate content of ß-branched amino acids (33% and 44%, respectively), are responsible for exocytosis of smaller-sized vesicles such as chromaffin granules (*Borisovska et al., 2005*), cytotoxic T-cell lytic granules (*Matti et al., 2013*) or small-synaptic vesicles (SSV) (*Schoch et al., 2001*), whereas syb1 with only 22% ß-branched amino acids preferentially mediates SSV exocytosis to release classical neurotransmitters at the NMJ (*Li et al., 1996*; *Liu et al., 2011*). Thus, the number of helix-destabilizing ß-branched amino acids within the N-terminal half of different v-SNARE TMDs appears to be evolutionary adapted to the size of vesicles to catalyze fusion pore expansion and facilitate bona fide cargo release. Such a mechanism could also tip the balance between an expanding or non-

**Table 1.** TMD sequence alignment of exocytotic v-SNARE variants. Amino acid residues comprising the putative TMD regions of the indicated v-SNARE variants are colored red. Note the different number and percentage of ß-branched amino acids (valine or isoleucine, bold) in the N-terminal half of the TMD as quantified on the right. Vesicle diameters are taken from the following references for small synaptic vesicles (*Takamori et al., 2006*), chromaffin granules (*Borisovska et al., 2005*), cytotoxic T-cell lytic granules (*Ming et al., 2015*), insulin granules (*Fava et al., 2012*), mast cell granules (*Alvarez de Toledo et al., 1993*), zymogen granules (*Nadelhaft, 1973*) and sequences were obtained from UniProt database.

| vesicle size | vesicle type (diameter) | v-SNARE isoform (M. musculus) | Transmembrane domain N term.   C term. | no. / % of V or I in the N-terminus |
|---|---|---|---|---|
| small | small synaptic vesicles (40 nm) | Synaptobrevin 1 | $^{93}$KNCK  MM**I**MLGA**I**C AIIVVVIVI YFFT$^{118}$ | 2 / 22 |
| inter-mediate | small synaptic vesicles (40 nm) chromaffin (120 nm), lytic (250 nm) and insulin (240 nm) granules | Cellubrevin | $^{78}$KNCK  MWA**I**G**I**S**V**L VIIVIIIIV WCVS$^{103}$ | 3 / 33 |
| | | Synaptobrevin 2 | $^{91}$KNLK  MM**II**LG**VI**C AIILIIIIV YFST$^{116}$ | 4 / 44 |
| large | mast cell and zymogen granules (500–800 nm) | VAMP7 | $^{185}$KNIKL**T**I**IIIIV**S**IV** FIYIIVSLLCGGFTW$^{215}$ | 8 / 73 |
| | | VAMP8 | $^{72}$KNVK  M**IVII**C**VIV** LIIVILIIL FATG$^{97}$ | 7 / 77 |

expanding fusion pore, on the one hand ensuring efficient discharge of bulky cargo molecules from large vesicles and on the other hand favoring release of small cargo as well as rapid recycling of SSVs by reducing the likelihood of complete merger with the plasma membrane.

Overall, our results unmask an active role of the proteinaceous TMD in membrane fusion that clearly goes beyond simple membrane anchoring and may be used to optimize release from differentially sized vesicles. ß-branched amino acids are key determinants for the fusogenic role of the v-SNARE TMD, most likely promoting the conformational dynamics of the TMD helix, which may perturb the packing of the surrounding phospholipids and thereby facilitate first intermembrane contact as well as fusion pore expansion. Taken together, SNARE proteins do not only act as force generators by continuous molecular straining on membranes, but also catalyze membrane merger via structural flexibility of their TMDs.

## Materials and methods

### Culture of chromaffin cells and hippocampal neurons

Experiments were performed on embryonic mouse chromaffin cells prepared at E17.5–E18.5 from double-v-SNARE knock-out mice (dko cells; Synaptobrevin-/-/Cellubrevin-/-, [*Borisovska et al., 2005*]) or syb2 knock-out mice (syb2 ko; Synaptobrevin-/- [*Schoch et al., 2001*]). Preparation of adrenal chromaffin cells was performed as described before (*Borisovska et al., 2005*). Recordings were done at room temperature on 1–3 days in culture (DIC) and 4.5–5.5 hr after infection of cells with virus particles.

Autaptic cultures of hippocampal neurons were prepared at E18 from syb2 knock-out mice, as described previously (*Bekkers and Stevens, 1991*; *Guzman et al., 2010*; *Schoch et al., 2001*). Recordings were performed at room temperature on days 11–15 of culture.

### Viral constructs

For expression in chromaffin cells, cDNAs encoding for syb2 and its TMD mutants were subcloned into the viral plasmid pSFV1 (Invitrogen, San Diego, CA), upstream of an internal ribosomal entry site (IRES) controlled open reading frame that encodes for enhanced green fluorescent protein (EGFP). EGFP expression (excitation wavelength 477 nm) was used as a reporter to identify infected cells. Mutant constructs were generated by PCR using the overlap expansion method (*Higuchi et al., 1988*). All mutations were confirmed by DNA sequence analysis (MWG Biotech, Germany). Virus cDNA was linearized with restriction enzyme SpeI and transcribed in vitro by using SP6 RNA polymerase (Ambion, USA). BHK21 cells were transfected by electroporation (400V, 975 μF) with a combination of 10 μg syb2 (wildtype/ mutant) and pSFV-helper2 RNA. After 15 hr incubation

(31°C, 5% $CO_2$), virions released into the supernatant were collected by low speed centrifugation (200 g, 5 min), snap-frozen and stored at -80°C (*Ashery et al., 1999*).

For transfection of neurons, cDNAs encoding for syb2 and its mutants were subcloned into pRRL. sin.cPPT.CMV.WPRE lentiviral transfer vector (*Follenzi et al., 2000*), which contains a cPPT sequence of the pol gene and the posttranscriptional regulatory element of woodchuck hepatitis virus (*Follenzi et al., 2002*). To identify transfected cells, syb2 proteins were expressed as fusion constructs with the monomeric red fluorescent protein (mRFP) linked to the C-terminal domain of syb2 via a 9aa linker (GGSGGSGGT). Mutant constructs were cloned analogous to the methods described above were verified by DNA sequence analysis. Lentiviral particles were produced as previously described (*Guzman et al., 2010*). Briefly, a 85% confluent 75 cm$^2$ flask of 293FTcells (Invitrogen) was transfected with 10 µg of the transfer vector, and 5 µg of each helper plasmid (pMDLg/pRRE, Addgene #12251; pRSV-Rev, Addgene #12253; pMD2.G, Addgene #12259) using a standard $CaCl_2$-$PO_4$ transfection protocol. Medium was exchanged 8 hr after transfection, viral particles were harvested after 48–72 hr, concentrated using a centrifugal device (100 kDa Molecular weight cutoff; Amicon Ultra-15; Millipore) and immediately frozen and stored at -80°C. Primary neurons were transfected with 300 µl of viral suspension (1DIC).

## Whole-cell capacitance measurements and amperometry of chromaffin cells

Whole-cell membrane capacitance measurements and photolysis of caged $Ca^{2+}$ as well as ratiometric measurements of $[Ca^{2+}]i$ were performed as described previously (*Borisovska et al., 2005*). The extracellular Ringer's solution contained (in mM): 130 NaCl, 4 KCl, 2 $CaCl_2$, 1 $MgCl_2$, 30 glucose, 10 HEPES-NaOH, pH 7.3, 320 mOsm. Ratiometric [Ca]i measurements were performed using a combination of fura2 and furaptra (Invitrogen) excited at 340 nm and 380 nm. The composition of the intracellular solution for flash experiments was (in mM): 110 Cs-glutamate, 8 NaCl, 3.5 $CaCl_2$, 5 NP-EGTA, 0.2 fura-2, 0.3 furaptra, 2 MgATP, 0.3 $Na_2GTP$, 40 HEPES-CsOH, pH 7.3, 310 mOsm. The flash-evoked capacitance response was approximated with the function: $f(x) = A0 + A1(1-exp[-t/t1]) + A2(1-exp[-t/t2]) + kt$, where A0 represents the cell capacitance before the flash. The parameters A1, t1, and A2, t2, represent the amplitudes and time constants of the rapidly releasable pool and the slowly releasable pool, respectively (*Rettig and Neher, 2002*). The stimulus-secretion delay was defined as the time between the flash and the intersection point of the back-extrapolated fast exponential with the baseline.

Production of carbon fiber electrode (5 µm diameter, Amoco) and amperometric recordings with an EPC7 amplifier (HEKA Elektronik) were done as described before (*Bruns, 2004*). For $Ca^{2+}$ infusion experiments, the pipette solution contained (in mM): 110 Cs-glutamate, 8 NaCl, 20 DPTA, 5 $CaCl_2$, 2 MgATP, 0.3 $Na_2GTP$, 40 HEPES-CsOH, pH 7.3, 310 mOsm (19 µM free calcium). Amperometric current signals were filtered at 2 kHz and digitized gap-free at 25 kHz. Amperometric events with a charge ranging from 10 to 5000 fC and peak amplitude >4 pA were selected for frequency analysis, while an amplitude criterion of >7 pA was set for the analysis of single spike characteristics. For fluctuation and rms noise analyses prespike signals with durations longer than 2 ms were considered and the current derivative was additionally filtered at 1.2 kHz. Fluctuations exceeding the threshold of ± 6 pA/ms (~4 times the average baseline noise) were counted. The number of suprathreshold current fluctuations divided by the corresponding prespike signal duration determines the fluctuation frequency.

## Electrophysiological measurements of synaptic currents

Whole-cell voltage-clamp recordings of synaptic currents were obtained from isolated autaptic or mass cultures of hippocampal neurons. All experiments include measurements from >3 different culture preparations and were performed on age-matched neurons derived from mice of the same litter. Intracellular solution contained (in mM): 137.5 K-gluconate, 11 NaCl, 2 MgATP, 0.2 $Na_2GTP$, 1.1 EGTA, 11 HEPES, 11 D-glucose, pH 7.3. Extracellular solution contained (in mM) 130 NaCl, 10 $NaHCO_3$, 2.4 KCl, 2 $CaCl_2$, 2 $MgCl_2$, 10 HEPES, 10 D-glucose, pH 7.3, 295 mOsm. To minimize the potential contribution of GABAergic currents the reversal potential of chloride-mediated currents was adjusted to the holding potential. Neurons were voltage-clamped at −70 mV (without correction for the liquid junction potential, $V_{LJ}$ 9.8 mV) with an EPC10 amplifier (HEKA Electronic) under

control of Pulse 8.5 program (HEKA Electronic) and stimulated by membrane depolarizations to +10 mV for 0.7 ms every 5 s (0.2 Hz). Cells with an average access resistance of 6–12 MΩ and with 70–80% resistance compensation were analyzed. Current signals were low-pass filtered at 2.9 kHz (four pole Bessel filter EPC10) and digitized at a rate of 10 or 50 kHz. The readily releasable pool (RRP) was determined by a 5 s application of hypertonic sucrose solution (500 mM sucrose) using a gravity-fed fast-flow system (*Bruns, 1998*). To accurately calculate the RRP size, the integral of current flow caused by a hypertonic solution was corrected by subtracting the amount of steady-state refilling and exocytosis that occurred during hypertonic challenges (*Stevens and Wesseling, 1999*). For recordings of spontaneous mEPSCs, mass cultures of hippocampal neurons were bathed in Ringer's solution containing 1 µM tetrodotoxin (TTX). To determine the mEPSC properties with reasonable fidelity events with a peak amplitude >15 pA (~5 times the S.D. of the background noise) and a charge criterion >25 fC were analyzed using a commercial software (Mini Analysis, Synaptosoft, Version 6.0.3).

## Biochemistry

SNAP-25 (amino acids 1–206) and Syntaxin 1a (amino acids 1–262) were expressed with an N-terminal 6-histidine tag ($His_6$) in the *E. coli* strain BL21DE3 and purified using nickel-nitrilotriacetic acid-agarose (Qiagen, Hilden, Germany). Recombinant variants of syb2 (amino acids 1–116) and syb2-polyL were expressed as N-terminal tagged GST fusion proteins (pGEX-KG-vector) in the *E. coli* strain BL21DE3 and purified using glutathione-agarose according to the manufacturer's instructions. All column elutes were analyzed for integrity and purity of the expressed proteins by SDS-PAGE and staining with Coomassie blue. SNARE complexes were formed by mixing equal molar amounts (~5 µM) of the proteins and incubating at 25°C for the indicated times (*Figure 1—figure supplement 1*). The binding buffer contained (in mM): 100 NaCl, 1 DTT, 1 EDTA, 0.5% Triton X-100, 20 Tris (pH 7.4). Assembly reactions were stopped by adding 5xSDS sample buffer. The ability of SNARE proteins to form SDS-resistant complexes was analyzed by SDS-PAGE (without boiling the samples) and Coomassie blue staining of protein bands.

## Immunocytochemistry

Chromaffin cells were processed 3.5 hr after virus infection for immunolabeling as described previously (*Borisovska et al., 2012*). An affinity purified mouse monoclonal antibody against syb2 (clone 69.1, antigen epitope amino acid position 1–14, kindly provided by R. Jahn, MPI for Biophysical Chemistry, Göttingen, Germany) and a rabbit polyclonal antibody against ceb (TG-21, synaptic system) were used for the immunocytochemical analysis. For epifluorescence microscopy, a Zeiss Axio-Vert 200 microscope was used, digital images (8 bit encoded) were acquired with a CCD camera and AxioVert Software (Zeiss, Germany) and analyzed with ImageJ software version 1.45. The total intensity of the fluorescent immunolabel was determined from (area of interest comprising the outer cell perimeter – area of interest comprising the cell nucleus).

To determine the localization and sorting of the mutant syb2 variants in large dense core vesicles, high resolution structured illumination microscopy (SIM) was employed. Cells were imaged through a 63x Plan-Apochromat (NA, 1.4) oil-immersion objective on the stage of a Zeiss Axio Observer with excitation light of 488 and 561 nm wavelengths. The ELYRA PS.1 system and ZEN software 2011 (Zeiss) were used for acquisition and processing of the images for SIM. Properties of syb2-fluorescent puncta in z-stacks were analyzed with the software package ImageJ, version 1.45. After threshold subtraction, Mander's weighted colocalization coefficients were determined from the sum of syb2 pixels intensities that colocalizes with ceb, divided by the overall sum of syb2 pixels intensities (*Bolte and Cordelieres, 2006*). Therefore $M_{Syb2} = \Sigma_{Syb}$ pixel intensity (coloc. ceb pixel)/ $\Sigma_{Syb}$ pixel intensity (*Manders et al., 1993*).

For immunostaining of the hippocampal neurons, cells were processed on 13 DIC as described for the chromaffin cells. Neurons were imaged with confocal microscope (LSM 510; Carl Zeiss) using the AxioVision 2008 software (Carl Zeiss) and a 100x, 1.3 NA oil objective at room temperature. Images were analyzed with the software package ImageJ (version 1.45) and SigmaPlot 8.0 (Systat Software, Inc.). Immunopositive spots were determined using a threshold-based detection routine, with the threshold adjusted to the background signal of the neuronal process. Immunosignals were

quantified as mean fluorescent intensity per puncta. For the analysis of synaptic density, synaptophysin-positive puncta were counted along 50 µm length of a neuronal process.

## Molecular dynamics simulations

The atomistic structure of the C-terminal region of syb2 (residues 71–116) was obtained from the X-ray crystallographic structure (*Stein et al., 2009*) and the missing C-terminal residue of syb2 (residue 116) was added using Modeller (*Sali and Blundell, 1993*). The insertion of the transmembrane domain of syb2 in an asymmetric bilayer was carried out using a self-assembly procedure described elsewhere (*Sharma et al., 2015*). Briefly, the atomistic structure was converted into a coarse-grained (CG) representation using a PerlScript file adapting the Martini coarse-graining method (*Monticelli et al., 2008*). The CG protein was positioned at the center of a box with dimensions of $10 \times 10 \times 11$ nm along with overlapping boxes of randomly placed cytoplasmic (CP) lipids and intravesicular (IV) lipids. The composition of CP lipids was 22 Palmitoyl-Oleoyl-PS (POPS), 76 Palmitoyl-Oleoyl-PE (POPE) and 92 cholesterol (CHOL) molecules and IV lipids was 22 Palmitoyl-sphingomyelin (PPCS), 66 Palmitoyl-Oleoyl-PC (POPC) and 52 CHOL. The resulting lipid box was filled with CG water and an appropriate number of $Na^+$ ions were added to preserve electro-neutrality. This was followed by 1000 steps of energy minimization using steepest descent algorithm after which ten production runs were carried out, each for 200 ns, using a time step of 2 fs. The effective time sampled in the production runs was therefore 2 µs. The CG simulations were analyzed for tilt angle of syb2 transmembrane domain and location of WW domain with respect to the phosphate groups of the membrane. The martini force field employs secondary structure constraints that do not allow changes in conformational states. Based on the CG analyses a representative structure was chosen and converted to an atomistic (AT) representation using a reverse transformation protocol (*Wassenaar et al., 2014*). Starting from the reverse-transformed atomistic wildtype structure of the syb2 C-terminal domain in the membrane, residues 97–112 were mutated to either Leu or to Val residues using Modeller to generate the respective mutants. The generated atomistic representations of syb2, polyL and polyV mutants were initially equilibrated for 2 ns with position restraints on the backbone heavy atoms using a harmonic force constant of 1000 kJ $mol^{-1}$ $nm^{-2}$. After this short equilibration, three 40 ns long production simulations were performed for the wildtype starting from different random velocities. For the mutants, two independent 40 ns long simulations were carried out. All analyses were done on the last 30 ns simulation, unless mentioned otherwise.

The lipid models used in the CG simulations, POPS, POPE, POPC, PPCS and the cholesterol model were CG Martini models and simulated using Martini force field ver. 2.0 and standard martini simulation parameters with a time step of 20 fs (*Monticelli et al., 2008*). The AT system was described using Slipids (Stockholm lipids) (*Jambeck and Lyubartsev, 2012*) for lipids, AMBER99SB-ILDN (ff99SB-ILDN) (*Lindorff-Larsen et al., 2010*) for protein and the waters were described using TIP3P (*Jorgensen et al., 1983*). A time step of 2 fs was used for AT simulations. All bonds were constrained using the LINCS algorithm. The bonds in water were constrained using the analytical SETTLE method (*Miyamoto and Kollman, 1992*). The pressure was kept constant at one atmosphere by a Parrinello-Rahman barostat (*Parrinello and Rahman, 1981*) with a coupling constant of 10.0 ps and an isothermal compressibility of $4.5 \times 10^{-5}$ $bar^{-1}$. A semi-isotropic coupling scheme was employed where the pressure in the xy plane (bilayer plane) is coupled separately from the z direction (bilayer normal). The Nosé-Hoover thermostat (*Hoover, 1985*; *Nose, 1984*) was used to maintain a constant temperature (323 K) with a coupling constant of 0.5 ps. Electrostatic interactions were calculated at every step with the particle-mesh Ewald method (*Essmann et al., 1995*) with real-space cutoff of 1.0 nm. The van der Waals interactions were cut off at 1.4 nm. All simulations were carried out with the GROMACS package version 4.6, (*Pronk et al., 2013*). Analyses were performed by using utilities within the GROMACS package. The secondary structure analyses were carried out using the dictionary of secondary structure of proteins (DSSP) method (*Kabsch and Sander, 1983*).

## Statistical analysis

Values are given as mean ± SEM (standard error of mean) unless noted otherwise in the figure legends. To determine statistically significant differences, one-way analysis of variance and a Tukey–Kramer post hoc test were used, if not stated otherwise.

## Acknowledgements

The authors would like to express their gratitude Drs. D Langosch, J Rettig, D Stevens and C Kummerow for valuable discussions. We thank W Frisch, P Schmidt, V Schmidt and M Wirth for excellent technical assistance and E Krause for help with the SIM-microscopy. The work was supported by grants from the DFG (SFB 1027 and GRK1326) to DB, and SFB 1027 and MO2312/1-1 to RM, and the ERC grant (ADG322699) to ML and by HOMFOR.

## Additional information

### Funding

| Funder | Grant reference number | Author |
|---|---|---|
| European Research Council | ADG322699 | Manfred Lindau |
| Deutsche Forschungsgemeinschaft | SFB1027 | Ralf Mohrmann<br>Dieter Bruns |
| Deutsche Forschungsgemeinschaft | GRK1326 | Ralf Mohrmann<br>Dieter Bruns |
| Deutsche Forschungsgemeinschaft | MO2312/1-1 | Ralf Mohrmann |
| Homburger Forschungsförderungsprogramm von Universitätsklinikum des Saarlandes | | Dieter Bruns |

The funders had no role in study design, data collection and interpretation, or the decision to submit the work for publication.

### Author contributions

MD, Performed experiments, Wrote the manuscript, Conception and design, Analysis and interpretation of data; AY, Performed experiments, Conception and design, Analysis and interpretation of data; MM, BS, YS, AS, SS, Performed experiments, Analysis and interpretation of data; RAB, Designed research, Acquisition of data, Analysis and interpretation of data; ML, Performed experiments, Designed research, Analysis and interpretation of data; RM, Designed research, Wrote the manuscript, Acquisition of data, Analysis and interpretation of data; DB, Performed experiments, Designed research, Wrote the manuscript, Analysis and interpretation of data

### Author ORCIDs

Rainer A Böckmann, http://orcid.org/0000-0002-9325-5162
Dieter Bruns, http://orcid.org/0000-0002-2497-1878

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
