## [Decision Letter]

Thank you for submitting your article "v-SNARE transmembrane domains function as catalysts for Ca^2+^-triggered vesicle fusion" for consideration by *eLife*. Your article has been favorably evaluated by Richard Aldrich as the Senior editor and three reviewers, one of whom is a member of our Board of Reviewing Editors. One of the three reviewers has agreed to reveal his identity: Matthijs Verhage (Reviewer #3).

The reviewers have discussed the reviews with one another and the Reviewing Editor has drafted this decision to help you prepare a revised submission. Generally, we are pleased to inform you that all three reviewers appreciate both the relevance of the questions asked and the high quality of the experimental work, and there is consensus that your work is acceptable for publication in *eLife* provided it is revised along the lines described below. Except for one (relatively minor) request for additional data (see below), all other comments affect your interpretation of the data and therefore (only) require a reply from you explaining your views and a description in which way the text has been revised.

Essential revisions:

Experimental:

The referees feel that information about unitary fusion events and/or spontaneous events should be provided. As a number of earlier studies have focused on this form of fusion, we think it is important to include this data in the manuscript in order to clarify how syb2 TM mutants alter the properties of single synaptic vesicle fusion events.

Editorial:

As you see below, referee 3 has raised several issues that we are asking you to consider and to respond to in your reply. In particular, the referees agree that the possible impact of interactions between synaptobrevn and syntaxin should be discussed, considering that such interactions were observed in the crystal structure of the complex.

Reviewer #2:

1) The title of the manuscript emphasizes the role of the syb2 transmembrane region in Ca^2+^-triggered exocytosis. Although most of the experiments assessed forms of release that were in fact Ca^2+^-dependent, the experiments conducted in autapses using hypertonic sucrose stimulation argue for a broader role, affecting fusion in general, both Ca^2+^-dependent and -independent forms of release. Therefore, I recommend removing "Ca^2+^-triggered" from the title.

Reviewer #3:

1) The authors do not sufficiently address the idea that SNARE-zippering is most likely a three step process in which the linker region at the membrane also interacts, as a last step. Stein et al. 2009 conclude that at least 10 amino acids interact with the Syntaxin linker/TMD, most of them in the syb TMD. This issue is insufficiently addressed. First, the experiments in Figure 1—figure supplement 1 do not address this issue critically. The results of these experiments are completely dominated by the interactions in the SNARE-domains, which are not altered and therefore really not expected to be changed (as the authors observed). This should be clearly indicated in the text. Second, the paper does not sufficiently address possible effects of Syb2 mutagenesis on interaction with the Syntaxin linker and TMD domains. When do the authors think these domains come together? Can the mutagenesis in Syb2 alter the previously predicted interactions? And can this help to explain differences in fusion pore opening? In the Discussion the authors conclude that "TMD mutations may change protein-lipid interactions during Ca^2+^-dependent fusion". What about protein-protein interactions? The Discussion mentions several more 'exotic' protein-protein interactions, such as proteinacious pore and homotypic interactions but not syb-syx. This should be discussed.

2) It is unclear why the poly-L mutant produces a strong impairment of secretion, in all phases of release, including the sustained phase that is considered to consist of vesicles not primed at the start of the stimulus. The kinetics of release are very similar, suggesting that the whole release process is scaled down or that the number of release sites is strongly reduced. There is no explanation for this important finding and it is not addressed in the Discussion.

3) Throughout the manuscript the authors state that the syb2 TMD plays "an active role" and even "catalyses" fusion. It is not clear what exactly the basis is for this statement. It seems likely that, since zippering starts at the N-terminals and then proceeds until the TMD, the TMD plays a rather passive role and simply follows the rest of the proteins. For the interpretation it seems not essential to make these claims. The authors should either provide clear experimental evidence/reasoning for this conclusion or adjust the conclusions.

4) The data using CSP anchors deviate from the data previously obtained by the Sudhof lab. The authors should acknowledge this and provide possible explanations.

---

## [Author Response]

Essential revisions:

*Experimental:*

*The referees feel that information about unitary fusion events and/or spontaneous events should be provided. As a number of earlier studies have focused on this form of fusion, we think it is important to include this data in the manuscript in order to clarify how syb2 TM mutants alter the properties of single synaptic vesicle fusion events.*

*Editorial:*

As you see below, referee 3 has raised several issues that we are asking you to consider and to respond to in your reply. In particular, the referees agree that the possible impact of interactions between synaptobrevn and syntaxin should be discussed, considering that such interactions were observed in the crystal structure of the complex.

As requested, we have now added results on quantal release with respect to frequency and amplitude in Figure 7—figure supplement 2. The text was changed accordingly:

“To test whether the flexibility of the syb2 TMD is also critical for quantal signaling, we recorded spontaneous excitatory postsynaptic currents (mEPSCs) in the presence of 1µM TTX using mass cultures of hippocampal neurons. […] Moreover, release from SSVs might be less dependent on TMD-mediated acceleration of fusion pore expansion due to the high curvature of the vesicle, as will be discussed below.”

*Reviewer #2:*

1) The title of the manuscript emphasizes the role of the syb2 transmembrane region in Ca^2+^-triggered exocytosis. Although most of the experiments assessed forms of release that were in fact Ca^2+^-dependent, the experiments conducted in autapses using hypertonic sucrose stimulation argue for a broader role, affecting fusion in general, both Ca^2+^-dependent and -independent forms of release. Therefore, I recommend removing "Ca^2+^-triggered" from the title.

We agree with the reviewer’s point that the syb2 TMD plays a general role in membrane fusion, affecting Ca^2+^-dependent as well as Ca^2+^-independent forms of release. In accordance with the reviewer’s suggestion, we have reformulated the title of the manuscript as *‘*v-SNARE transmembrane domains function as catalysts for vesicle fusion’.

*Reviewer #3:*

*1) The authors do not sufficiently address the idea that SNARE-zippering is most likely a three step process in which the linker region at the membrane also interacts, as a last step. Stein et al. 2009 conclude that at least 10 amino acids interact with the Syntaxin linker/TMD, most of them in the syb TMD. This issue is insufficiently addressed. First, the experiments in Figure 1—figure supplement 1 do not address this issue critically. The results of these experiments are completely dominated by the interactions in the SNARE-domains, which are not altered and therefore really not expected to be changed (as the authors observed). This should be clearly indicated in the text.*

While SNARE complex assembly (Figure 1—figure supplement 1) is largely mediated by the cognate SNARE motifs, previous studies have also shown that the TMD of v-SNAREs changes stability to the SNARE complex (Stein et al., 2009). This prompted us to examine whether the loss of function phenotype seen in polyL mutant is due to altered SNARE complex formation, a scenario that may serve as potential explanation for the observed reduction of the flash evoked response. To further clarify this point, we now rephrased our statement:

“By studying SNARE complex assembly with recombinant proteins, we found that the polyL variant affects neither the rate nor the extent of SNARE complex formation (Figure 1—figure supplement 1).[…] Thus, the secretion deficiency in polyL expressing cells is not due to impaired SNARE complex formation, i.e. by causing changes in vesicle priming, but rather reflects defective vesicle fusion.”

Second, the paper does not sufficiently address possible effects of Syb2 mutagenesis on interaction with the Syntaxin linker and TMD domains. When do the authors think these domains come together? Can the mutagenesis in Syb2 alter the previously predicted interactions? And can this help to explain differences in fusion pore opening? In the Discussion the authors conclude that "TMD mutations may change protein-lipid interactions during Ca^2+^-dependent fusion". What about protein-protein interactions? The Discussion mentions several more 'exotic' protein-protein interactions, such as proteinacious pore and homotypic interactions but not syb-syx. This should be discussed.

In the original manuscript we have discussed a potential interference with heteromeric interactions between TMDs of Syntaxin and Synaptobrevin. Still, we fully agree with the reviewer’s suggestion that this issue requires a more elaborated discussion. Furthermore, we also included a discussion of phenotypic differences between TMD mutants and previously measured mutations in the juxtamembrane domain of syb2 (Borisovska et al., 2012). Accordingly, we have rephrased and extended the text:

“In this context it is noteworthy, that neither deletion nor substitutions of membrane-proximal tryptophane (Trp) residues within the juxtamembrane domain (JMD) of syb2 (Borisovska et al., 2012) were found to alter the tonic secretion response or fusion pore properties as observed here with the TMD mutants (Figure 2 and Figure 5). […] Thus, heterodimerization between v- and t-SNARE TMDs likely succeeds but does not promote fusion pore opening and expansion.”

2) It is unclear why the poly-L mutant produces a strong impairment of secretion, in all phases of release, including the sustained phase that is considered to consist of vesicles not primed at the start of the stimulus. The kinetics of release are very similar, suggesting that the whole release process is scaled down or that the number of release sites is strongly reduced. There is no explanation for this important finding and it is not addressed in the Discussion.

Our findings with the polyL mutant show a strong impairment of secretion at all phases of release, which we attributed to altered fusogenic properties of the syb2 protein, as discussed in the second paragraph of the Discussion section. Alternatively, one might suggest that the TMD mutants may somehow change the ‘priming’ reaction upstream of the final fusion event. However, as outlined already under point #1) and described in the first paragraph of the subsection “Stabilization of the syb2 TMD helix diminishes synchronous secretion”, we find no evidence of impaired SNARE complex formation with polyL mutant. In contrast, our MD simulations provide significant evidence for altered conformational flexibility of the polyL TMD that can be held responsible for the reduced fusion activity of the protein (Figure 4). In good agreement, previous NMR-studies have shown that increasing the content of ß-branched amino acids of TMD mimic peptides profoundly enhanced lipid mobility and wobbling of lipid head groups (Agrawal et al., 2010; Agrawal et al., 2007), which could lower the energy barrier for fusion. Furthermore, our results are supported by previous in vitro work, showing that isolated SNARE TMDs (Langosch et al., 2001) in the absence of any cytoplasmic domain of syb2 facilitate liposome-liposome fusion. Overall, these data provide strong evidence that structural features of v-SNARE TMDs are crucial for Ca^2+^-triggered exocytosis, enabling TMDs to actively promote the fusion process.

As described above, the reviewer also suggests that the number of release site may be reduced in polyL mutant expressing cells. While this hypothesis cannot be rigorously excluded, to best of our knowledge we are not aware about results from the literature supporting such a scenario. Indeed, it is difficult to envision how TMD mutant variants of a vesicular protein may alter the number of release sites on the plasma membrane.

3) Throughout the manuscript the authors state that the syb2 TMD plays "an active role" and even "catalyses" fusion. It is not clear what exactly the basis is for this statement. It seems likely that, since zippering starts at the N-terminals and then proceeds until the TMD, the TMD plays a rather passive role and simply follows the rest of the proteins. For the interpretation it seems not essential to make these claims. The authors should either provide clear experimental evidence/reasoning for this conclusion or adjust the conclusions.

Our functional analyses clearly show that substitution of amino acids with helix stabilizing leucine residues specifically in the N-terminal half of the TMD strongly reduces both, overall secretion and fusion pore dynamics (Figure 1, Figure 2, Figure 3, Figure 5 and Figure 5—figure supplement 1). This secretion deficit indicates an active, fusion promoting role of the syb2 TMD and has been verified by independent measurements of membrane capacitance and catecholamine secretion (amperometry). This conclusion is further substantiated by the experimental evidence outlined under point #2). In addition, a similar reduction in exocytosis and slowing of fusion pore kinetics was observed by replacing the syb2 TMD with an acylated syb2-CSP fusion protein confirming that conformational flexibility of a proteinaceous v-SNARE TMD is required for fusion. Moreover, an unprecedented gain of function phenotype accelerating the kinetics of fusion pore expansion beyond the rate of wildtype cells was seen with three different TMD mutants (polyV, polyI and polyI-Nt mutants). This further underlines an active, fusion promoting role of the syb2 TMD that clearly goes beyond simple membrane anchoring. These results and possible mechanisms by which the syb2 TMD promotes fusion have been discussed in detail in the Discussion section.

*4) The data using CSP anchors deviate from the data previously obtained by the Sudhof lab. The authors should acknowledge this and provide possible explanations.*

As requested, we have now included a paragraph providing an explanation for the apparent differences between our data obtained with the acylated syb2-CSP fusion protein in chromaffin cells and the previously reported results by Zhou et al., 2013 in neurons:

“Our data obtained with the acylated syb2-CSP fusion protein appear to deviate from the previously reported results by Zhou et al., 2013, wherein the mutant protein significantly rescued synaptic transmission compared to a syb2-RST-mVenus construct serving as control. […] Consequently, the reduced ability of the syb2-RST-mVenus construct to rescue neuronal exocytosis may have led to an overestimation of the acylated syb2-CSP response providing an explanation for the apparently discrepant results.”